# External mechanical loading overrules cell-cell mechanical communication in sprouting angiogenesis during early bone regeneration

Chiara Dazzi[1], Julia Mehl[1], Mounir Benamar[2], Holger Gerhardt[3,4,5], Petra Knaus[2], Georg N. Duda[1,6], Sara Checa[1]*

1 Julius Wolff Institute for Biomechanics and Musculoskeletal Regeneration, Berlin Institute of Health at Charité–Universitätsmedizin Berlin, Berlin, Germany, 2 Institute for Chemistry and Biochemistry, Freie Universität Berlin, Berlin, Germany, 3 Max Delbrück Center for Molecular Medicine, Berlin, Germany, 4 DZHK (German Centre for Cardiovascular Research), Partner Site Berlin, Berlin, Germany, 5 Berlin Institute of Health at Charité–Universitätsmedizin Berlin, Berlin, Germany, 6 Berlin Institute of Health Centre for Regenerative Therapies (BCRT), Berlin Institute of Health at Charité–Universitätsmedizin Berlin, Berlin, Germany

* sara.checa@bih-charite.de

## Abstract

Sprouting angiogenesis plays a key role during bone regeneration. For example, insufficient early revascularization of the injured site can lead to delayed or non-healing. During sprouting, endothelial cells are known to be mechano-sensitive and respond to local mechanical stimuli. Endothelial cells interact and communicate mechanically with their surroundings, such as outer-vascular stromal cells, through cell-induced traction forces. In addition, external physiological loads act at the healing site, resulting in tissue deformations and impacting cellular arrangements. How these two distinct mechanical cues (cell-induced and external) impact angiogenesis and sprout patterning in early bone healing remains however largely unknown. Therefore, the aim of this study was to investigate the relative role of externally applied and cell-induced mechanical signals in driving sprout patterning at the onset of bone healing. To investigate cellular self-organisation in early bone healing, an *in silico* model accounting for the mechano-regulation of sprouting angiogenesis and stromal cell organization was developed. Computer model predictions were compared to *in vivo* experiments of a mouse osteotomy model stabilized with a rigid or a semirigid fixation system. We found that the magnitude and orientation of principal strains within the healing region can explain experimentally observed sprout patterning, under both fixation conditions. Furthermore, upon simulating the selective inhibition of either cell-induced or externally applied mechanical cues, external mechanical signals appear to overrule the mechanical communication acting on a cell-cell interaction level. Such findings illustrate the relevance of external mechanical signals over the local cell-mediated mechanical cues and could be used in the design of fracture treatment strategies for bone regeneration.

**Data Availability Statement:** All relevant data are within the manuscript and its Supporting Information files. The agent-based model codes and finite element model input files are publicly available on GitHub at https://github.com/dazzich/Angio_early_bone_healing.

**Funding:** This study was funded by the German Research Foundation (DFG) through the Collaborative Research Centre (CRC) 1444, project ID: 427826188 (CD, JM, MB, PK, HG, GD, SC). The funders had no role in study design, data collection and analysis, decision to publish, or preparation of the manuscript.

**Competing interests:** The authors have declared that no competing interests exist.

## Author summary

After bone injury, the timely growth of new blood vessels from existing ones—a process known as sprouting angiogenesis—is essential for a proper bone healing process. Although we know that sprouting angiogenesis is influenced by mechanical cues, how cell-induced forces or external physiological loads interact to drive sprouting angiogenesis during the early stages of bone healing remains largely unknown. To investigate this, we developed a computational framework that simulates sprouting angiogenesis during the early healing phase, since the different mechanical cues are difficult to isolate and investigate experimentally. Through the comparison of computer model predictions with the experimental data, we identified the magnitude and direction of the principal strains within the healing zone as key drivers of vessel invasion and patterning. After confirming that our computer model correctly captured vessel and cell organization during early bone healing, we performed *in silico* experiments to better understand the relative contribution of the various mechanical cues to the sprouting process. We found that externally applied mechanical loads overrule the cell-cell mechanical communication. Our findings suggest that external mechanical loads could be taken into consideration to design more effective treatment strategies for the regeneration of bone. This study demonstrates the importance of combining *in silico* and experimental techniques to gain new insights behind complex mechano-biological processes such as the mechanical regulation of sprouting angiogenesis.

## Introduction

Sprouting angiogenesis—the process by which new blood vessels emerge from existing vasculature—plays a major role during many physiological and pathological processes [1], including bone regeneration. In a bone injury, blood vessels that cross the fracture line are disrupted. To achieve healing, the vascular network needs to be re-established to enable oxygen, nutrients and growth factor supply. Angiogenesis usually starts during the first week post-fracture [2] and this initial phase is critical for the healing outcome. A lack or inhibition of angiogenesis results in delayed healing or even non-union [3–5]. Non-unions are characterized by a lower density of vessels during the first week post-surgery, while vascularization reaches values comparable to successful healing cases at later time points, suggesting a pivotal role of early sprouting angiogenesis for the ultimate healing outcome [6,7]. Thus, understanding the mechanisms involved during early angiogenesis is of crucial importance to develop effective treatments.

Both biochemical [8] and mechanical signals [9] are known to collectively drive sprouting angiogenesis. While the role of biochemical signals in sprouting angiogenesis during the early healing phase has been intensively investigated [5,10,11], the role of mechanical cues remains poorly understood. During sprouting, endothelial cells (ECs) are known to be sensitive and generate specific responses to local mechanical signals [9], like mechanical strains arising within the extracellular matrix (ECM). The mechanical environment perceived by ECs within the healing region is highly dynamic and influenced by the cells themselves which exert appreciable traction forces on the ECM during migration. Similarly, outer-vascular stromal cells (OVSCs, heterogeneous class of cells that make up the connective tissue, eg. fibroblasts, pericytes, mesenchymal stromal cells [12]), pull on the ECM during cell migration and sense and respond to substrate deformation [13]. When co-cultured with ECs, such stromal cells exhibit a supporting role for vessel organization [14] since ECs failed to organize into vessels in the absence of outer-vascular cells. Mechanical signals have been shown to be involved in the

interaction between OVSCs and ECs. Inhibition of fibroblast force transmission and traction forces resulted in a damaged vascular network [15], while an increase in fibroblast traction forces promoted sprout formation [16]. These experimental observations suggest that a complex mechanical interplay exists between ECs and OVSCs via the ECM deformation; however, their role in angiogenesis during early bone healing remains largely unknown.

On a macroscopic level, tissues are constantly deformed as a consequence of physiological activity. External load-induced mechanical strains are transmitted to the ECM and impact cell responses. Several experimental studies have investigated the effect of externally applied boundary conditions on ECs and OVSCs organization. Neo-vessel sprouting, elongation and alignment during angiogenesis depend not only on the magnitude and frequency of the external load [17] but also on the loading mode (cyclic vs. static) [15]. Similarly, OVSCs are influenced by local mechanical signals: specifically, fibroblast-like cells migrate towards stiffer regions (*durotaxis phenomenon*) [18] and, in cyclically stressed environments, re-orient in order to avoid substrate deformations [19,20]. In the clinical scenario of bone healing, mechanical strains within the ECM are determined not only by the patient's physical activity, but also by the gap size and the mechanical stability of the fracture fixation system. In the specific context of early bone healing, however, it remains so far unknown how these dynamic mechanical signals acting at different scales impact cell self-organization and sprout patterning.

Mechanical signals are challenging to measure and to investigate experimentally. Multiple mechanical cues likely have synergistic effects on cells, which are difficult to analyse independently. *In silico* models that are validated against *ex vivo* data offer a powerful tool to analyse mechanical signals acting at different length scales as well as researching their effect independently and isolating the distinct impacts of various mechanical cues.

Over the last decades, computer models have allowed gaining a better understanding of the underlying mechanisms driving angiogenesis (reviewed elsewhere [21–23]). Only a few computer models have focused on the interaction between the outer-vascular mechanics and the angiogenic process [24–27]. In the specific context of bone healing, a description of angiogenesis is included in several computer models in order to: (i) incorporate the effect of oxygen supply by the vascular network [28–31] and therefore have a more realistic prediction of the bone regeneration outcome; (ii) investigate the impact of impaired angiogenesis on fracture repair [32, 33]; (iii) evaluate the effects of scaffold architecture on angiogenesis [34–36]. Despite the prominent role of cell-matrix mechanical interactions, only a few computational models of sprouting angiogenesis explicitly deal with cell traction forces [26,37]. Van Oers and colleagues [37] developed a mathematical model of angiogenesis taking into account traction forces generated by cells and the matrix mechanics. They demonstrated that cell traction force-induced matrix strain is important for multicellular organization, such as during sprouting angiogenesis. However, neither the effect of external loads nor the presence of OVSCs was taken into account.

This study aims to fill this gap by investigating the relative role of externally applied loads (e.g. gait load) and cell-induced traction forces on sprout patterning during the early stages of bone healing, using a computer modelling approach. To achieve this, *in silico* models of the mechano-regulation of OVSCs and ECs organization during early bone healing were developed. As a first step, mechano-biological rules regarding sprout patterning and stromal cell organization were derived from the experimental literature and implemented in the early bone healing models. Computer model predictions were then compared with *ex vivo* experimental data of cell and sprout patterning 7 days post-surgery in a mouse osteotomy model. Thereafter, the models were used to perform *in silico* experiments where the distinct mechanical components (cell-induced and external) were removed selectively.

## Materials and methods

### Ethics statement

All experimental animal procedures were reviewed and approved by the local animal protection authority (Landesamt für Gesundheit und Soziales, LaGeSo, approval number: G0322/18) and were performed in accordance with the German Animal Welfare Act.

### Experimental setups

Eight female mice aged 9–10 weeks underwent a diaphyseal osteotomy on the left femur stabilized by an external rigid (n = 4) or semirigid (n = 4) fixation system (MouseExFix 100% and 50%, RISystem, S1 Fig). Surgery was performed under anaesthesia with an isoflurane-oxygen mixture. An incision through the skin was made from the knee to the hip joint. The iliotibial tract and vastus lateralis were dissected and the femur was exposed. Four holes were drilled into the femur and the fixator was mounted. A 0.7-mm osteotomy was performed using a wire saw (RISystem, Davos, Switzerland) between inner screws. After surgery, animals were administered 25 mg tramadol per mL in the drinking water for 3 days. Mice were sacrificed 7 days post-osteotomy and femurs were carefully harvested and fixed in 4% paraformaldehyde for 6–8 hours under agitation. The bone samples were then washed and decalcified in EDTA solution (24h at 4˚C under agitation), which was followed by incubation in sucrose solution (20% sucrose, 2% PVP) (6h at 4˚C under agitation). Bones were transferred into a cryosection mould, mounted with embedding bone medium (8% gelatin, 20% sucrose, 2% PVP) and stored at -80˚C. To identify vessels within the healing region and quantify microvascular network parameters, 50 μm-thick cryosections were prepared for immunofluorescence staining. Cryosections were thawed and dried for 30 min at room temperature, rehydrated with ice-cold PBS and permeabilized with ice-cold 0.3% triton in water. Subsequently, sample sections were stained with an antibody against Endomucin (Emcn) (Santa Cruz, Cat# sc-65495, 1:100), diluted in 5% normal serum donkey in PBS, and incubated overnight at 4˚C. The primary antibody was stained with the secondary antibody after washing the samples with ice-cold PBS for 3 x 5 min. Samples were finally washed with PBS and mounted with Fluoromount G (#Cat 0100–01, Southern Biotech).

### *In silico* models

*In silico* multi-scale models of the mechano-regulation of sprouting angiogenesis and OVSCs organization within a bone healing region were developed, replicating the experimental set-up. Here, mouse osteotomies stabilized with either a rigid or a semirigid external fixator were used as *in vivo* experiments. Finite Element Models (FEMs) were generated at the tissue scale to compute the mechanical strains within the healing region. These were iteratively coupled to Agent-Based Models (ABMs) simulating cellular activity and allowing to investigate the ECs and OVSCs response to local mechanical signals (e.g. *durotaxis*) (Fig 1). After the initialization step, that represents the OVSCs and ECs seeding within the ABM, at every iteration, cellular traction forces are assigned to OVSCs and tip ECs based on their current position and orientation within the healing region and are given as input to the FEM in the form of concentrated loads. The finite element analysis is executed automatically and the predicted principal strains and local deformations are read by the ABM. According to a set of rules describing the mechano-regulation of OVSCs and ECs organization that are detailed in the following paragraphs, OVSCs and ECs position and orientation are updated within the ABM and the algorithm starts over (Fig 1). The iterative nature of the two model layers allowed obtaining results at discrete time points with each iteration corresponding to a subsequent process of healing. In

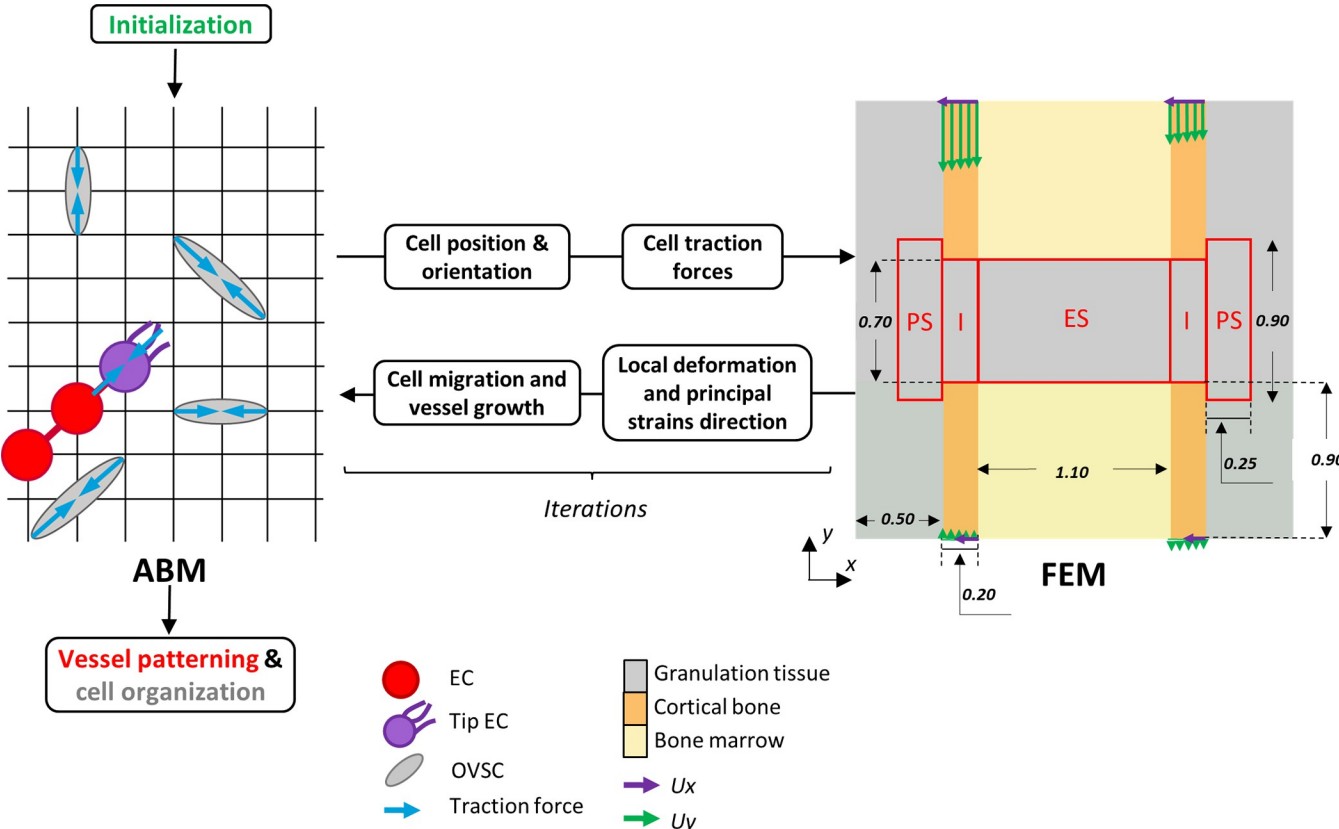

**Fig 1. Schematic representation of the coupling between ABMs and FEMs.** All measures are expressed in mm. Red boxes identify the regions of interest for strain distribution prediction and are identical to Borgiani et al. (2019) [43] to facilitate the comparison between the models. PS = periosteal, I = intracortical, ES = endosteal. Purple and green arrows represent displacement boundary conditions applied to the cortices along x and y, respectively.

our modelling approach, we assumed that an iteration represents a time step of 20 minutes in a cascade of mouse osteotomy healing. Simulations were performed so as to replicate the first week post-fracture. The predicted sprout patterning on day 7 was compared to the *in vivo* experimental results at the same time point. OVSCs organization on day 7 was analysed in terms of cell orientations and compared to available experimental observations of collagen organization during the early stages of bone healing. To our knowledge, there are no experimental studies that have quantified OVSCs organization during the early stages of bone healing. However, collagen fibres have been reported to relate well with stromal cell orientation [38,39]. On the one hand, cells were shown to remodel and align the surrounding ECM by exerting traction forces [40,41]. On the other hand, aligned ECM fibres guide cellular orientation and migration [42].

## Finite element models at the tissue scale

FEMs replicating the healing region of the *in vivo* experimental setups were built through the commercial finite element solver Abaqus (Abaqus 3DEXPERIENCE R2019x). The mid-longitudinal section of the healing region was modelled including the cortical bone ends, the marrow cavity and the osteotomy gap (Fig 1). The models were meshed through a regular grid of 4 nodes plane strain elements (CPE4) with a distance between adjacent nodes of 0.01 mm. Linear elastic material properties were assigned to the tissues comprised within the fracture, i.e.

**Table 1. ECM material properties and cell activities parameters.**

| Parameter (Unit) | | Value |
|---|---|---|
| *Granulation tissue* | Young's modulus (MPa) | 0.2 [a] |
| | Poisson's ratio | 0.167 [a] |
| *Bone marrow* | Young's modulus (MPa) | 2 [b] |
| | Poisson's ratio | 0.167 [a] |
| *Cortical bone* | Young's modulus (MPa) | 5000 [c] |
| | Poisson's ratio | 0.3 [a] |
| Vessel rate of growth (μm/h) | | 15 [d] |
| Probability of tip EC migration in the direction of the previous iteration [P1] | | 0.4 [*] |
| Probability of tip EC migration in a random direction [P2] | | 0.4 [*] |
| Probability of tip EC migration following the strain-based rule [P3] | | 0.2 [*] |
| OVSCs migration rate (μm/day) | | 30 [e] |
| OVSCs proliferation rate (%/day) | | 33 [e] |
| OVSCs apoptosis rate (%/day) | | 0.03 [e] |
| ECs traction force total magnitude (N) | | $20 \times 10^{-6}$ [f] |
| OVSCs traction force total magnitude (N) | | $32 \times 10^{-6}$ [g] |

[a][45]
[b][46,47]
[c][48]
[d][49]
[e]Adapted from [43]
[f]Adapted from [50]
[g]Adapted from [51]
[*]estimated through a parameter sweep analysis (S2 File).

granulation tissue, bone marrow and cortical bone [44] (Table 1). The presence of the external fixator, for both a rigid and semi-rigid fixation system, was virtually simulated using corresponding boundary conditions. Specifically, a previously developed and validated 3D finite element model of a 0.5 mm mouse osteotomy stabilized with the same rigid and semi-rigid fixators and subjected to external physiological loading [43] was adapted to replicate the 0.7 mm defect induced in the current study (S2 Fig). The displacement profiles of the cortices were recorded along the two orthogonal axis of the mid-longitudinal plane and applied as displacement boundary conditions (Ux, Uy, Fig 1) to the cortical bone fragments of the 2D model. A mesh convergence study was conducted for the 3D finite element model and resulted in an optimal average characteristic element size of 0.1 mm, one order of magnitude larger than the one adopted in the 2D model. At each iteration, concentrated loads representing traction forces exerted by cells onto the matrix were given as input to the FEMs.

## Agent-based models at the cellular scale

ABMs, consisting of several algorithms in C++, were built to describe cell behaviour. The models included two types of agents (i.e. cells): OVSCs and ECs. The specific mechano-biological rules guiding OVSCs organization and vessel growth are described in the following paragraphs. Assuming an average neo-vessel and OVSCs diameter of 0.01 mm [52], coordinates of finite element nodes were used to create the 2D lattice grid of potential positions for cells. Therefore, the distance between lattice points was equal to 0.01 mm.

## Modelling sprouting angiogenesis

Microvessels were modelled as a sequence of ECs occupying subsequent lattice points. ECs were initialized on 10% and 1% of lattice points within the periosteum and bone marrow, respectively. This choice was motivated by the observed dominant vascular response from the periosteum [53,54] and the diminished vessel ingrowth from the medullary cavity [55] during early bone healing. The leading tip ECs were simulated to migrate to a newly available position, while following stalk ECs occupy the positions left empty. The leading tip ECs of each sprout were assumed to behave like an active force dipole oriented along the sprout growth direction. Two concentrated forces (traction forces) were applied onto the tip ECs adjacent nodes, directed along the dipole direction and pointing towards the cell core [56]. The selected configuration of EC traction forces is based on 3D traction force microscopy data of small sprouts (2–3 ECs) that was used to compute the traction forces directions leading to the experimentally determined matrix deformation (S1 File) [57,58]. Stalk ECs were assumed to exert no traction since negligible ECM deformations have been observed experimentally near stalk cells as compared to those generated close to the sprout tips [57,58]. Traction force total magnitude for tip ECs (Table 1) was obtained from the curve reported in [50] by assuming a linear relationship with the substrate elastic modulus. Vessel growth direction was determined by the leading tip ECs (known to probe the environment through filopodia and respond to guidance cues by leading vessel growth [59]) based on three options with assigned probabilities (Pn, n = 1,2,3) (Table 1 and Fig 2): at each iteration, a tip EC can migrate towards the direction of the previous iteration (P1), towards a random direction (P2) or following a strain-based rule (P3 = 1-P2-P1). To estimate the values of these probabilities, they were changed systematically through a parameter sweep analysis. The combination of values that better matched experimental data in terms of sprout patterning was selected and is reported in Table 1. More details on the parameter sweep analysis are provided in S2 File.

This strain-based rule was established based on experimental observations. Shear strains are known to be detrimental to angiogenesis [60] and therefore, it was assumed that vessels grow along the absolute maximum principal strain directions ($EP_{max\_abs}$), where the shear strains are 0. Several studies report that ECs and vessel fragments cyclically stretched at 10% [17,61,62], 15% [63] or 20% [64] orient themselves perpendicular to the direction of the $EP_{max\_abs}$ as a structural response to minimize the stress experienced by the cells. In the developed model, we assumed that tip ECs migrate along the direction of the $EP_{max\_abs}$ until a certain strain value (5%), identified as the upper limit strain value favourable for bone formation [65]. It was hypothesized that ECs gradually avoid the direction of the $EP_{max\_abs}$ for higher magnitudes and orient themselves fully perpendicular to the $EP_{max\_abs}$ direction for strains above 10%. Furthermore, tip ECs stop migrating if the value of $EP_{max\_abs}$ is above 30%, according to the reduced vascularity observed under cyclic compressive strains higher than 30% at the early stage of healing [66] (Fig 2).

Algorithms for branching and anastomosis were adapted from Checa & Prendergast, 2009 [28]. In brief, sprout formation by branching out from a parent vessel was modelled as a stochastic process where the probability of a sprout to form from a vessel segment is proportional to the segment length. When the leading tip EC attempts to cross the path of another sprout, or its own path, anastomosis took place and the EC lose its tip phenotype.

To account for the real 3D geometry of the callus and generate more realistic results despite the 2D simplification, we included the possibility for tip ECs to virtually go to and come from out of the plane. Every 15 iterations, considering the persistence time in a plane [34], all the potential positions surrounding an agent in a 3D grid were considered (7 in the plane and 10 out of the plane) and vessels growing along a random direction had 10/17 possibilities to

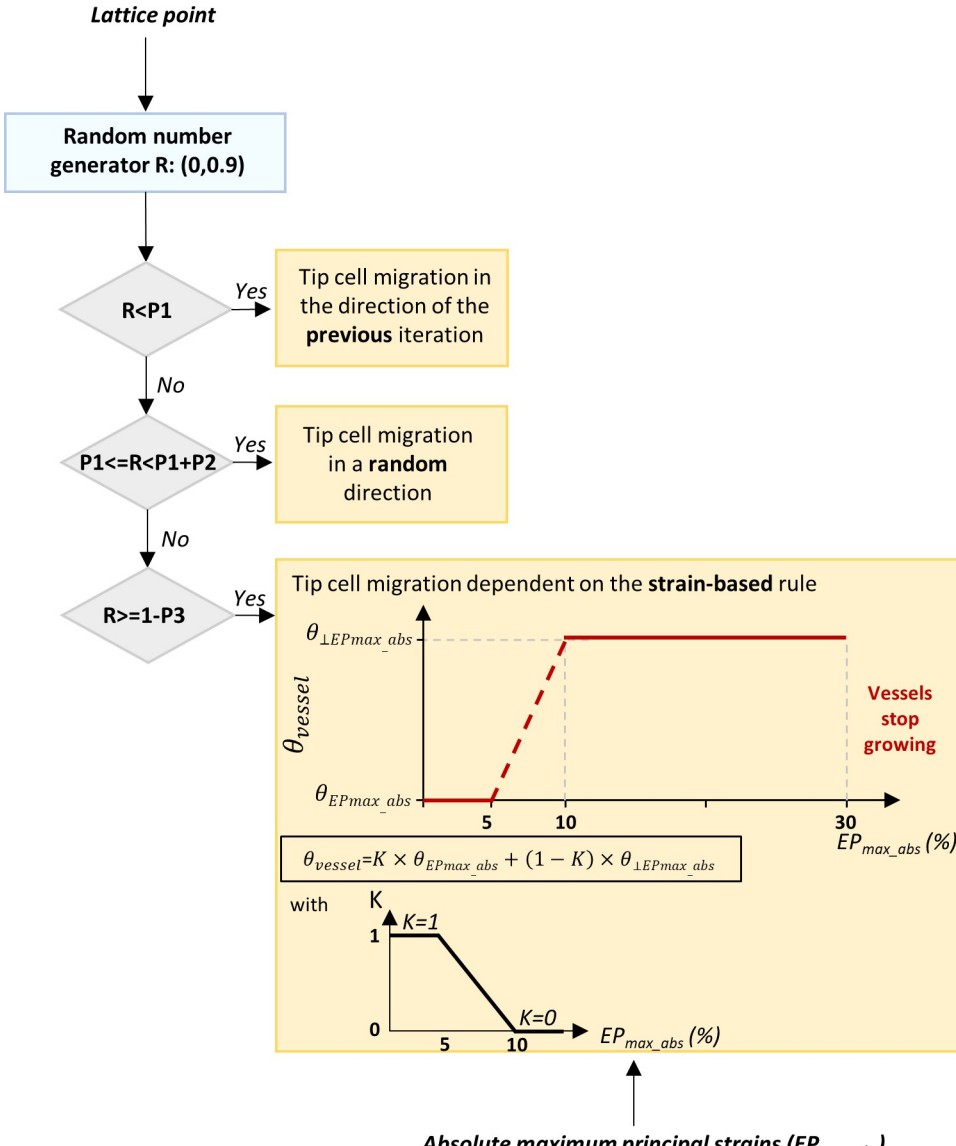

**Fig 2. Algorithm implemented to simulate vessel growth.** P1 = probability of tip EC migration in the direction of the previous iteration; P2 = probability of tip EC migration in a random direction; P3 = probability of tip EC migration following the strain-based rules (P3 = 1-P2-P1); $EP_{max\_abs}$ = absolute maximum principal strain; $\theta_{vessel}$ = vessel growth direction; $\theta_{EPmax\_abs}$ = absolute maximum principal strain direction; $\theta_{\perp EPmax\_abs}$ = perpendicular to the absolute maximum principal strain direction.

disappear from the plane. Assuming that each section adjacent to our model experienced the same, tip ECs were allowed to enter the simulated domain from the out-of-plane direction with the number of vessels coming equal to the number of vessels going out. Moreover, to replicate the gradual penetration of vessels from the external soft tissues, the perimeter of a virtual 4mm-diameter circumference representing the callus [43] was initialized with ECs. Throughout the simulation, the virtual circumference radius decreased its length at a speed equal to the vessel rate of growth and lattice points external to the circumference were progressively allowed to host new ECs (with an estimated probability of 0.5%) provided that the ECs density in the surrounding was below 10% [2].

## Modelling stromal cells activity

At the beginning of the simulation, 20% of lattice points within the bone marrow and periosteum were assumed to be filled with OVSCs [67]. OVSCs were represented as active force dipoles [68,69] exerting two concentrated loads, as described above for ECs. The same linear relationship between the substrate elastic modulus and the traction force magnitude adopted for ECs [50] was assumed for OVSCs. The total force magnitude exerted by fibroblasts on a softer material [51] was therefore scaled accordingly (Table 1). OVSCs could assume 4 possible orientations in the plane: horizontal, vertical, diagonal left and diagonal right. During the first days post-surgery, the release of growth factors at the healing site, like Transforming Growth Factor Beta [70], is known to attract OVSCs. To simulate this directed migration of OVSCs towards the osteotomy gap, OVSCs migration was modelled as a random walk with a preference for less densely populated locations during the first 4 days post-surgery. In addition, OVSCs have been reported to probe the local deformation along the dipole direction [71] and to change their position and dipole orientation towards areas of highest stiffness (*durotaxis*) [18]. Therefore, after 4 days, OVSCs migration was biased by a rule dependent on local stiffness, based on Checa et al. (2015) [72]. Shortly, at each iteration, cells migrate randomly to one of their free neighbouring positions (8 possibilities) where they adopt a random preferential direction. In the new configuration, the cell measures the local substrate deformation by applying traction forces and either adopts the new configuration (if the deformation in the new position is lower) or not. In case no available positions are present surrounding a migrating cell, the cell does not migrate (contact inhibition). In addition, OVSCs proliferation and apoptosis were included, whose rates were adapted from Borgiani at al. (2019) [43] taking into account the 2D simplification.

Whenever possible, parameters were taken from experimental data. A list of all parameter values used in the model is provided in Table 1, including references.

## *In silico* experiments

After the comparison of model predictions with experimental data, further *in silico* experiments were carried out to achieve a more mechanistic understanding of the relative contribution of each mechanical signal to sprout patterning as well as the consequences of pathological alterations in cell mechano-response. To achieve this, mechanical cues were selectively removed by adapting the model as indicated in the following paragraphs. From here on, we will refer to the mechano-biological rules explained above as "Baseline".

## Inhibition of tip ECs mechano-response (EC_MR_KO)

A reduced cellular mechano-response has been already proposed as the responsible mechanism for delayed bone regeneration in some clinical scenarios, like ageing [43]. How an altered cell mechano-response affects sprout formation remains unclear. We hypothesize that the response of tip ECs to mechanical signals plays a fundamental role during early sprouting angiogenesis. To assess this, we performed *in silico* experiments where tip ECs did not respond to the local mechanical signals, i.e. P3 was set equal to 0. The tip ECs migration probabilities P1 and P2 were adjusted to account for the reduced organization of vessel structures observed experimentally upon inhibition of cell mechano-responders, such as ROCK, under cyclic stretch [64]: P1 (persistency) was kept at 0.4, while P2 (randomness) was raised to 0.6.

### Inhibition of OVSCs traction forces (OVSC_TF_KO)

To better understand the contribution of cell-induced and external mechanics for sprout patterning, traction forces exerted by OVSCs were knocked out. OVSCs forces were set equal to 0 N. Simulations were then compared to the baseline model.

### Unloaded condition

Unloading is known to be disruptive to bone regeneration [73,74]. However, to isolate the contribution of the cell-induced mechanical signals, the boundary conditions at the cortices were removed and the bone cortices were fixed in all degrees of freedom. As a control, OVSCs traction forces were inhibited under unloading conditions and simulation results in the presence and absence of OVSCs traction forces were compared.

### Quantification of microvascular network parameters

Due to the stochastic component included in the initial seeding and in the migration rules for both ECs and OVSCs, six realizations for each simulation were performed. The realization closest to the mean is reported as representative image for each simulated scenario. *In silico* and *ex vivo* results are reported as mean ± standard deviation.

The vessel density was computed within a few regions of interests (ROIs) distinguishing between the gap, the bone marrow, and the periosteum for both *in silico* and *ex vivo* data through the Fiji plugin "Vessel analysis" [75]. In particular, the vessel length density ratio was extracted, that is the ratio of the skeletonized area to total area within the ROI. This allowed disregarding the influence of vessel cross-section that is constant in the computer model.

Image analysis of vessel orientation distribution was performed through the Fiji plugin "Directionality" within the same ROIs for both *in silico* and *ex vivo* data. The plugin generates a histogram indicating the percentage of structures (vessel elements) in a given direction, spanning from 0˚ to 180˚. We then grouped the percentage of structures into 20˚-width bins (S3 Fig). To emphasize the preferential orientation in each ROI, if any, the average percentages of vessels aligned along the bone long axis direction (80˚-100˚ bin), the direction perpendicular to the bone axis (0˚-20˚ and 160˚-180˚) and all the other directions were compared.

Vessel morphometric differences between the baseline and the EC_MR_KO models were quantified by computing the distribution of vessel lengths and the number of self-intersecting vessels that generate closed loops.

### Statistics

Experimental data were tested for normal distribution using the Shapiro-Wilk normality test. The statistical significance of the vessel length density ratio between either different ROIs or experimental and computational results within the same ROI was assessed through a two-tail student t-test in Matlab (R2020b).

The statistical significance of the number of vessel elements aligned along a preferred orientation as compared to all other directions was assessed within each ROI both for *in silico* and *ex vivo* results. Due to the non-normal distribution of certain dataset, the Kruskal-Wallis H test was carried out to compare n>2 groups. In case a significant difference between groups was detected, the post-hoc non-parametric Mann-Whitney U-test was performed in Matlab (R2020b) for pairwise comparisons complemented by Bonferroni correction. The level of significance was defined as $p \leq 0.05$.

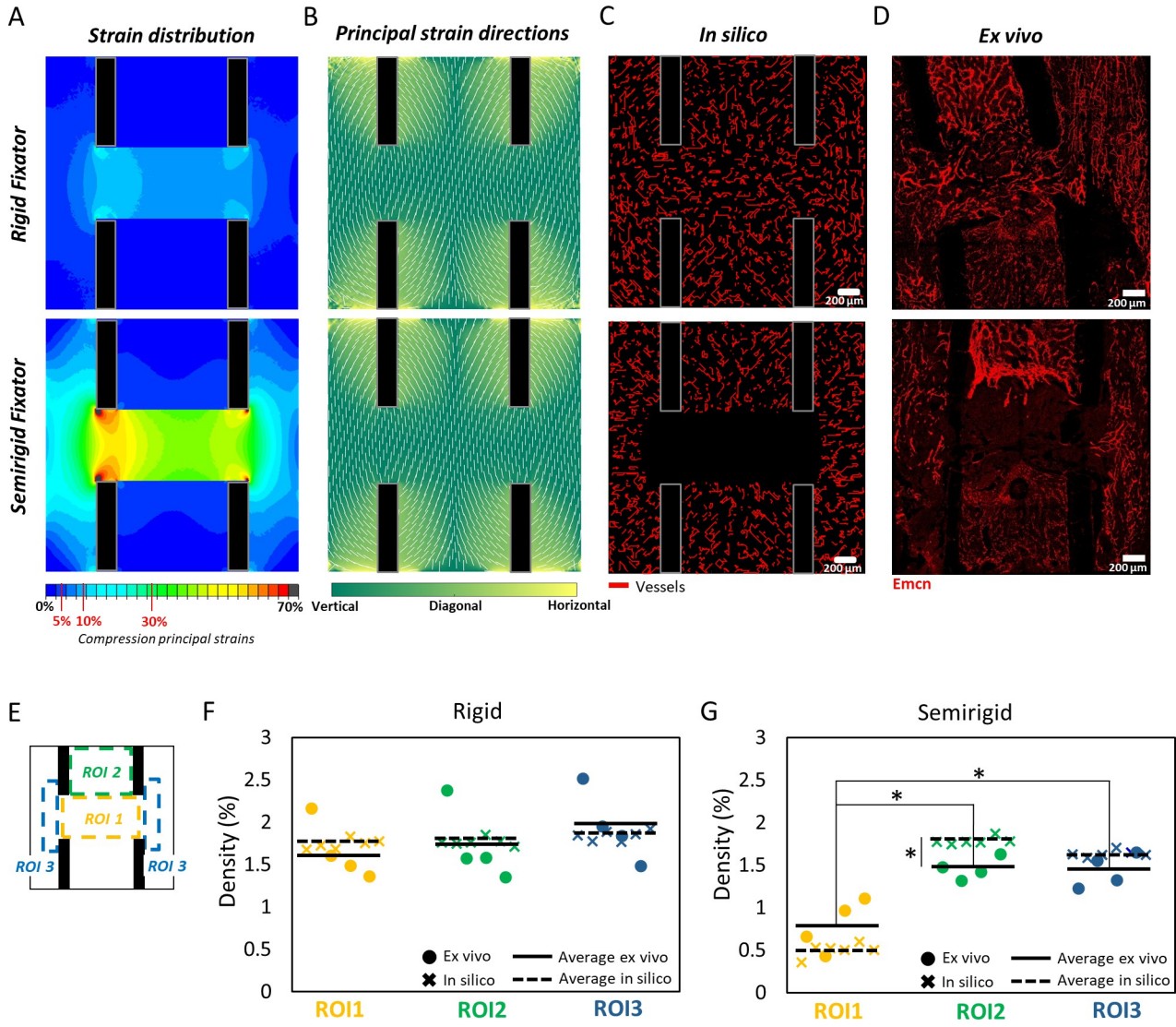

**Fig 3. Predictions of mechanical strains, vessel patterning and density, and comparison with experimental data.** On the top, the healing region on the 7th day post-osteotomy under rigid (top row) and semirigid (bottom row) fixation conditions; (A) predicted strain distribution; (B) principal strain directions; (C) *in silico* predictions of vessels pattern; (D) *ex vivo* vessels pattern (Emcn, Endomucin stained); (E) regions of interest (ROIs); (F-G) scatter plot of experimental vs. predicted vessel density in ROIs under rigid vs. semirigid fixation conditions. Circles represent experimental samples; crosses represent the *in silico* realizations; solid lines indicate the average experimental vessel density; dashed lines indicate the average *in silico* vessel density. Asterisks indicate a significant difference: *P value < 0.05, **P value < 0.01.

## Results

### Mechanical strains and principal strain directions can explain vessel patterning at the onset of healing

On the 7th day post-surgery, average absolute maximum principal strains (compressive strains) within the endosteal gap were predicted to be -11.1±0.9% and -44.8±3.9% in the case of rigid and semirigid fixators, respectively. Higher strains were predicted intercortically on the medial side (rigid fixator: -13.9±1.5%; semirigid fixator: -57.1±6.2%) compared to the lateral side, i.e. fixator side (rigid fixator: -11.7±1.2%; semirigid fixator: -46±4.7%), under both fixator conditions. The periosteal region exhibited lower compressive strains compared to the intercortical region but

larger strains were found medially (rigid fixator: -7.9±2.2%, semirigid fixator: -32.5±9%) compared to laterally (rigid fixator: -6.2±1.7%, semirigid fixator: -23.9±6.7%) also on the periosteal side (Fig 3A). Absolute maximum principal strains were mainly aligned with the bone long-axis, in line with the orientation of the imposed displacement boundary conditions at the cortices (Fig 3B).

Qualitatively, computer model predictions of vessel patterning 7 days post-surgery agreed with experimental observations: under rigid fixation, vessel segments were predicted across the whole osteotomy gap region while under semirigid fixation, vessels were not predicted within the endosteal gap (Fig 3C and 3D). Also, predicted vessel densities within different regions of interest (Fig 3E) agreed with experimental data (Fig 3F–3G). Under semirigid fixation, the model was capable of mimicking the lack of vascularity obtained *in vivo* within the gap (ROI 1) that is reflected by the significantly (P value < 0.05) reduced vessel density (Fig 3G) and can be related to the high strain values locally, above 30%. However, under semirigid fixation, the model overestimated (P value < 0.05) vessel density in ROI 2 (bone marrow). This discrepancy can be due to the constant rate of growth assigned to vessels that simplifies the complexity of the biological system.

Besides vessel density, *in silico* predictions appeared to replicate the preferential alignment of newly formed sprouts observed experimentally across the different ROIs (Fig 4). Both *in silico* and *ex vivo*, under rigid fixation, a preferential orientation along the bone long axis (90˚) was found within the bone marrow (ROI 2) and in the periosteal region (ROI 3), in agreement with the direction of compressive principal strains. As vessels started approaching the osteotomy gap, strains rose and led to a gradual reorientation of vessels towards the horizontal or 0˚ direction (Fig 4). Under the semirigid fixation, vessels were predicted to organize themselves within the bone marrow (ROI 2) similarly to the rigid case, with a preference for the 90˚ orientation (bone long axis). However, periosteally (ROI 3), vessels were predicted to take a preferential horizontal orientation due to the high strains measured locally (above 20%), which was not observed experimentally (Fig 4). Although *in silico* results did not match exactly the experimental ones in terms of average percentage of vessel structures aligned in a particular direction, the model could capture the significant preferential alignment observed experimentally. By comparing the *in silico* vs. *ex vivo* normalized curves of the distribution of vessel orientations across the ROIs (S4 Fig) it appears clear that the preferential alignment compares well, apart from the periosteal site (ROI 3) in the semirigid fixator case. This discrepancy is likely due to the influence of other factors, such as the haptotactic cues provided by ECM fibres, neglected in the model. The original plots displaying the percentage of vessel structures for each 20˚-width bin are shown in S3 Fig.

A movie of vessel invasion over the first week of healing is provided in S1 Movie and S2 Movie files for the rigid and semirigid fixator respectively.

### *Durotaxis* can explain early OVSCs organisation during bone healing

Due to the lack of experimental data on OVSCs organization during the early stages of bone healing, OVSCs alignment was compared to the experimentally observed alignment of collagen fibres during the early healing phase (Fig 5A). It has already been shown that ECM fibres align well with cellular orientation [38,39]. Indeed, cells are known to constantly remodel and align their ECM by exerting traction forces [40,41] and, in turn, aligned ECM fibres influence cellular orientation and migration [42]. The comparison to experimental data was carried out in two mechanically diverse regions of interest, where collagen fibres were found preferentially aligned along the surfaces of the cortical bone (Fig 5A): at the bone periosteum and within the osteotomy gap. Similarly, *in silico* predictions showed that around 70% of OVSCs aligned perpendicular to the bone long axis within the gap while periosteally nearly 50% of OVSCs aligned along the long bone axis direction (Fig 5B and 5C).

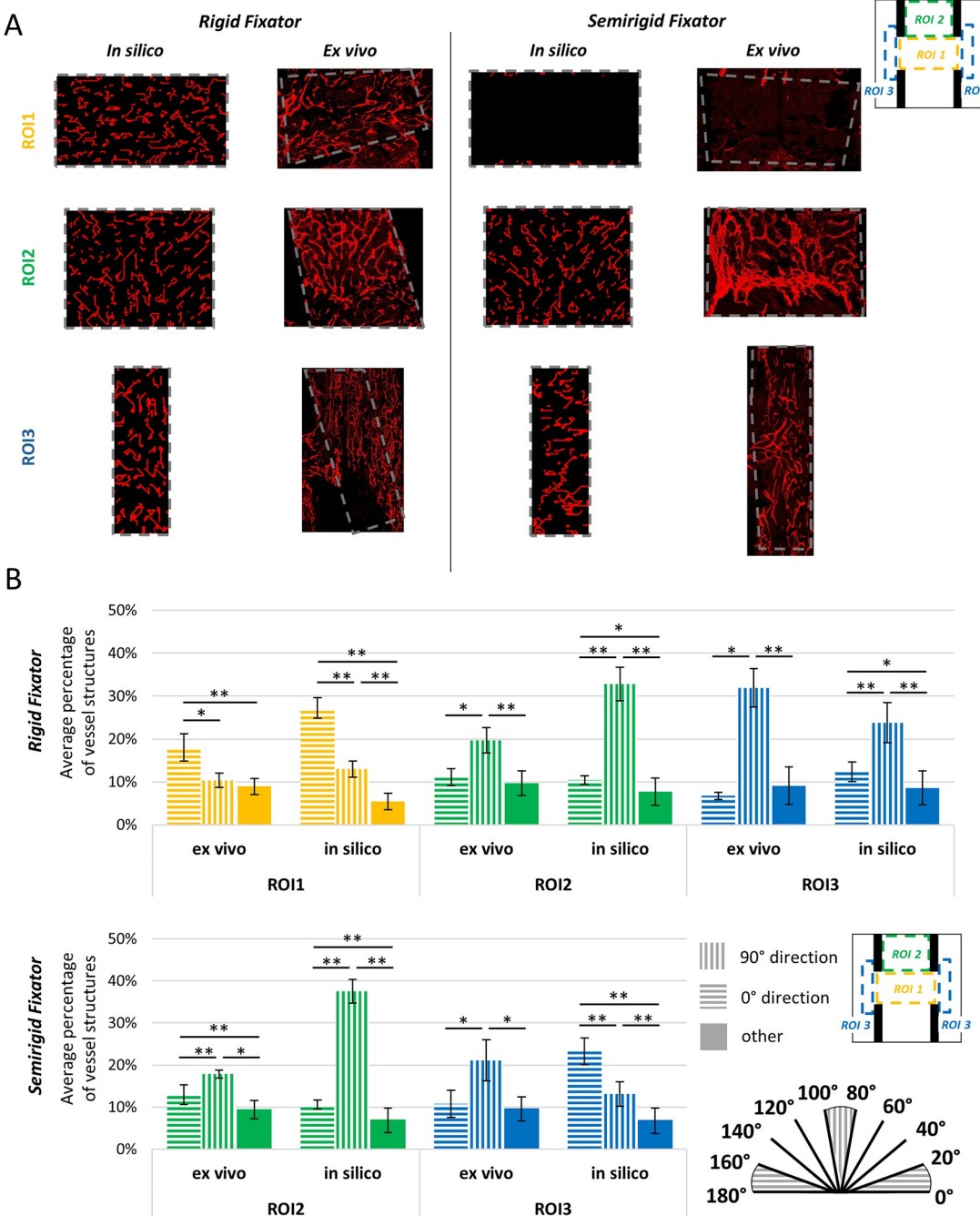

**Fig 4. Qualitative and quantitative comparison of the predicted vessel orientation with experimental data in the different ROIs.** A) *In silico* vs. *ex vivo* comparison between the zoomed ROIs (from the top, ROI 1, ROI 2, ROI 3) for each fixator type (rigid on the left, semirigid on the right); (B) Vessel orientation analysis results for the rigid (top) and semirigid (bottom) fixators. The colours represent the ROIs, the patterns represent the preferred direction: 90°, 0°, other. Asterisks indicate a significant difference: *P value < 0.05, **P value < 0.01.

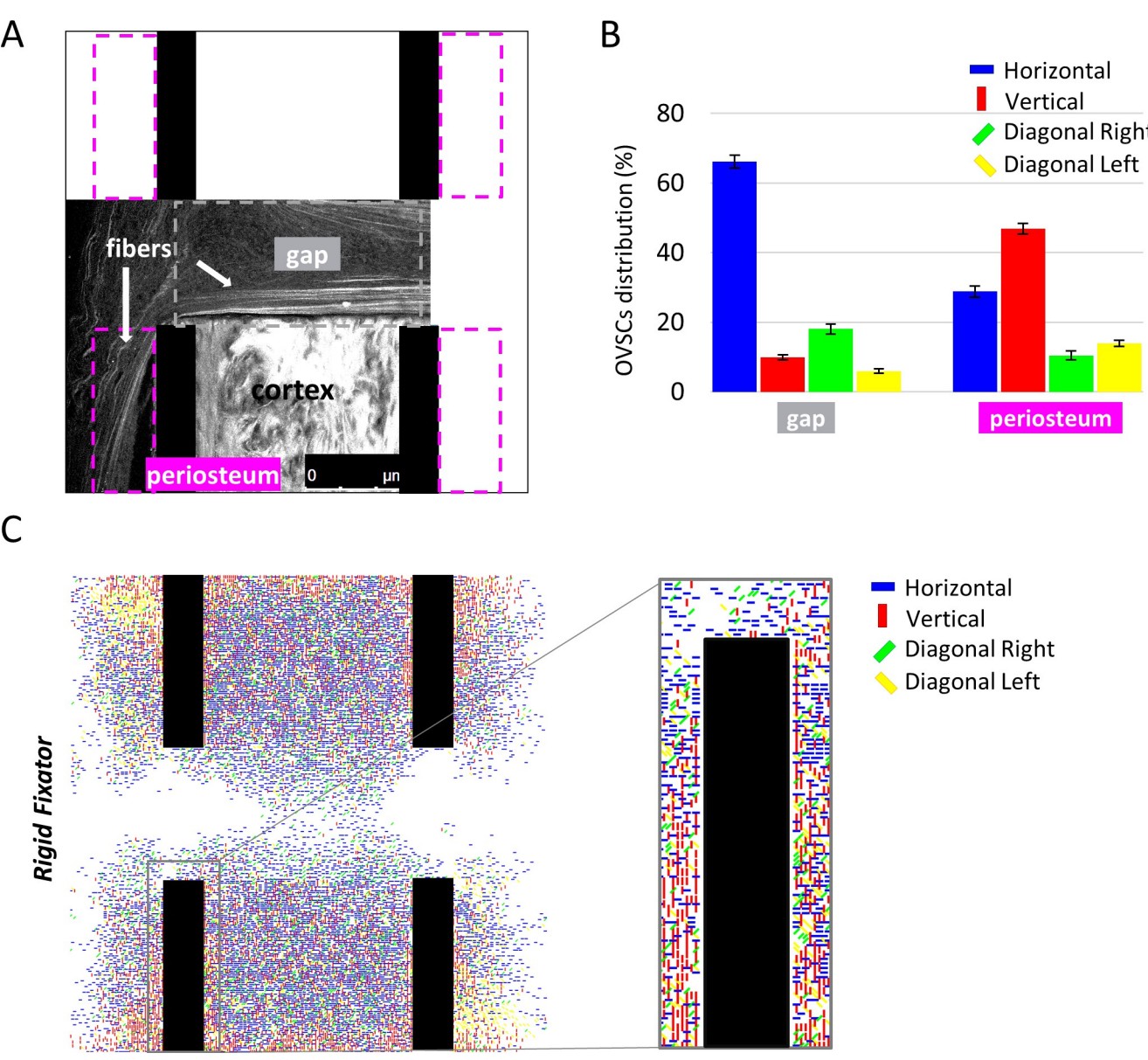

**Fig 5. Predicted OVSCs organization under rigid fixation and comparison with experimental data.** (A) Alignment of newly formed collagen fibres (white arrows) in the proximity of the cortex (second harmonic signal, two-photon microscopy) during the initial healing phase. The regions of interest considered in the analysis are identified with dashed lines; (B) Percentage distribution of OVSCs orientation for the rigid fixator within the gap and along the periosteum. On the top right, the colour code for cell orientation is reported; (C) Predicted OVSCs self-organization at day 7 for the rigid fixator. On the top right, the colour code for cell orientation is reported.

The corresponding images for the semirigid fixator are reported in S5 Fig, given the similarity of the results.

## Tip cell mechano-responsiveness appears to be key to vessel growth

With the rigid fixator, inhibition of tip EC mechano-responsiveness led to an altered vessel organization (Fig 6A). While vessels mainly aligned horizontally within the gap in the baseline case, no remarkable preferential orientation was observed when tip EC mechano-response was removed in the model (Fig 6B). To confirm this observation, the angle bins in which a higher

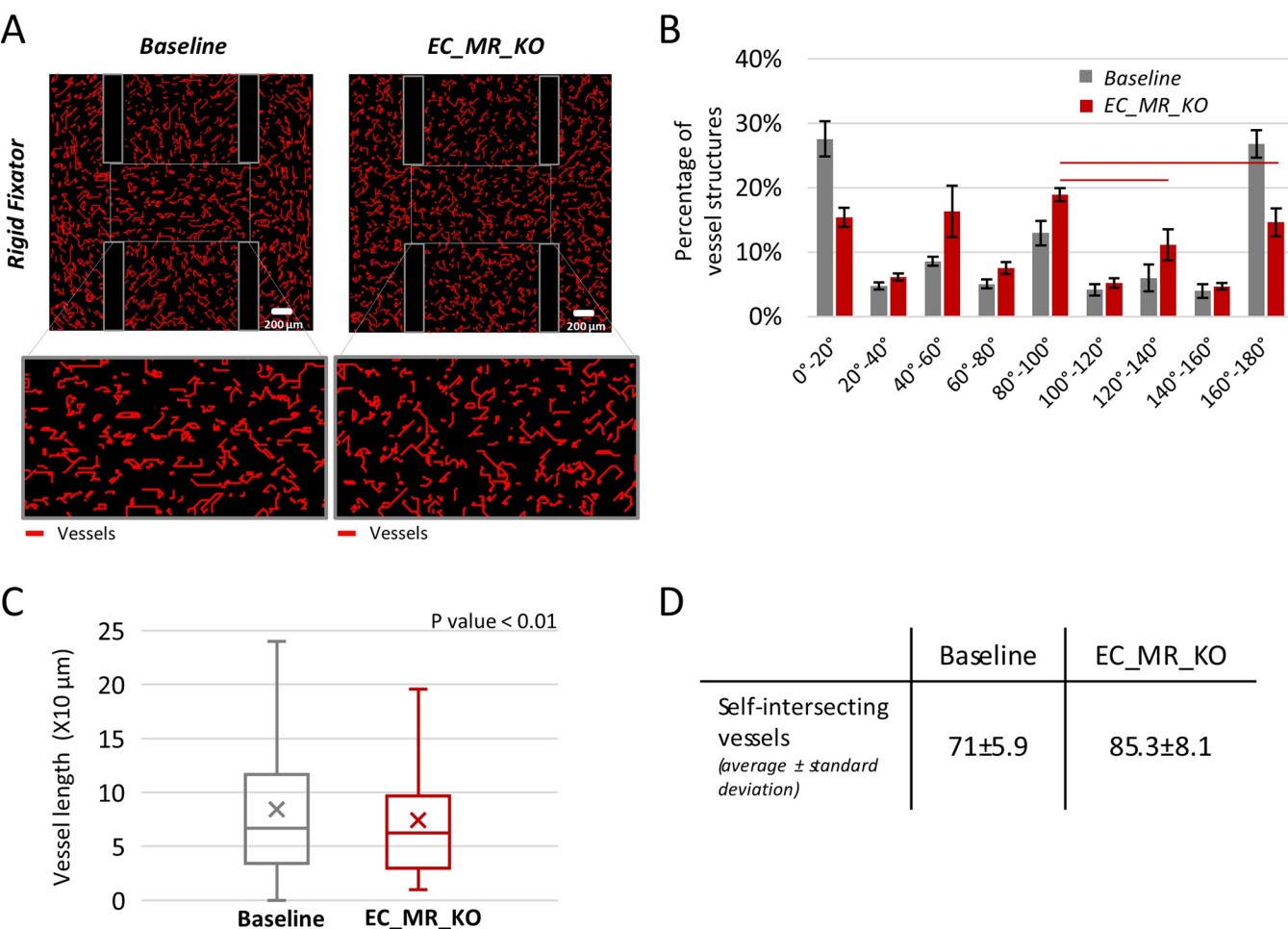

**Fig 6. Predicted vessel organization in the baseline model vs. upon endothelial cells mechano-response inhibition.** (A) Predicted vessel distribution for the baseline (left) and EC_MR_KO (right). The zoomed images of the osteotomy gap are reported below; (B) vessel orientation analysis for EC_MR_KO and baseline within the gap. Lines indicate a significant difference in the pairwise comparisons between angle bins in the EC_MR_KO model (p<0.05); (C) Box plots of vessel lengths for the baseline and EC_MR_KO within the whole healing region. The average vessel length is significantly different between the two groups (p<0.01); (D) The table shows the average number of self-intersecting vessels for the baseline and EC_MR_KO within the whole healing region. EC_MR_KO = inhibition of tip ECs mechano-response.

percentage of vessel structures was detected (0˚-20˚,40˚-60˚,80˚-100˚,120˚-140˚,160˚-180˚) were compared statistically and only the following pairs were determined to be significantly different: 80˚-100˚ compared to 120˚-140˚ and 80˚-100˚ compared to 160˚-180˚. Moreover, under mechano-response inhibition, vessel fragments were characterized by shape defects. Specifically, the length of vessels decreased under mechano-response inhibition compared to the baseline case (Fig 6C), from an average length of 85 μm to 75 μm. This can be explained by the increased number of vessels that create closed loops by intersecting their own path (Fig 6D), and therefore stop elongating.

## Increased external loads overrule mechanical interactions on a cell-cell level

To better understand the role of OVSCs-induced traction forces on sprouting, we removed traction forces exerted by OVSCs and investigated the perturbation induced in terms of vessel patterning. Interestingly, vessel organization was not affected by traction force inhibition in

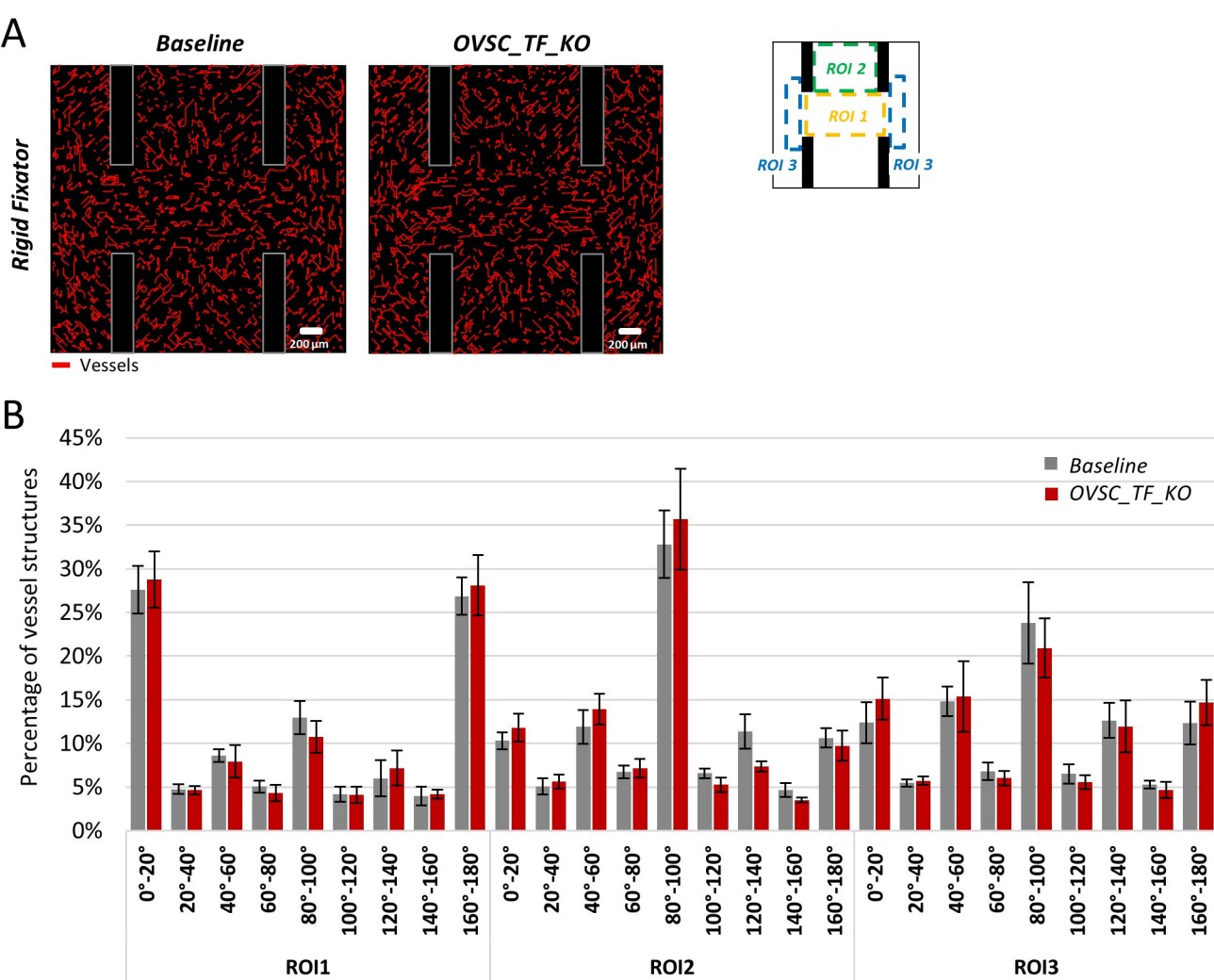

**Fig 7. Predicted vessel organization in the baseline model vs. upon stromal cell traction forces inhibition.** (A) Predicted vessel distribution for the baseline (left) and OVSCs_TF_KO (right); (B) vessel orientation analysis for the baseline vs. OVSCs_TF_KO model within the ROIs. OVSCs_TF_KO = inhibition of OVSCs traction forces.

OVSCs (Fig 7A). The orientation analysis revealed an analogous distribution of vessel directions for both the baseline and the OVSC_TF_KO models in all the regions of interest (Fig 7B) and over each angle interval.

## Cell-cell mechanical interaction emerges after unloading the osteotomy

Motivated by the lack of influence of OVSCs traction force inhibition on vessel patterning, we next attempted to isolate the effect of cell forces from the external mechanics by unloading the cortices and imposing a fully restrained motion in all six degrees of freedom.

In the unloaded model, strains were induced by cell contractility only and their pattern reflect the OVSCs organization. As expected, absolute maximum principal strains within the healing region were reduced as compared to the loaded cases, below 2% (Fig 8A). Cellular traction forces induced mainly compressive strains within the callus, but tensile strains were predicted within the core of the gap (Fig 8A). Within the gap, principal strain directions were found to be aligned along the bone long-axis (Fig 8B).

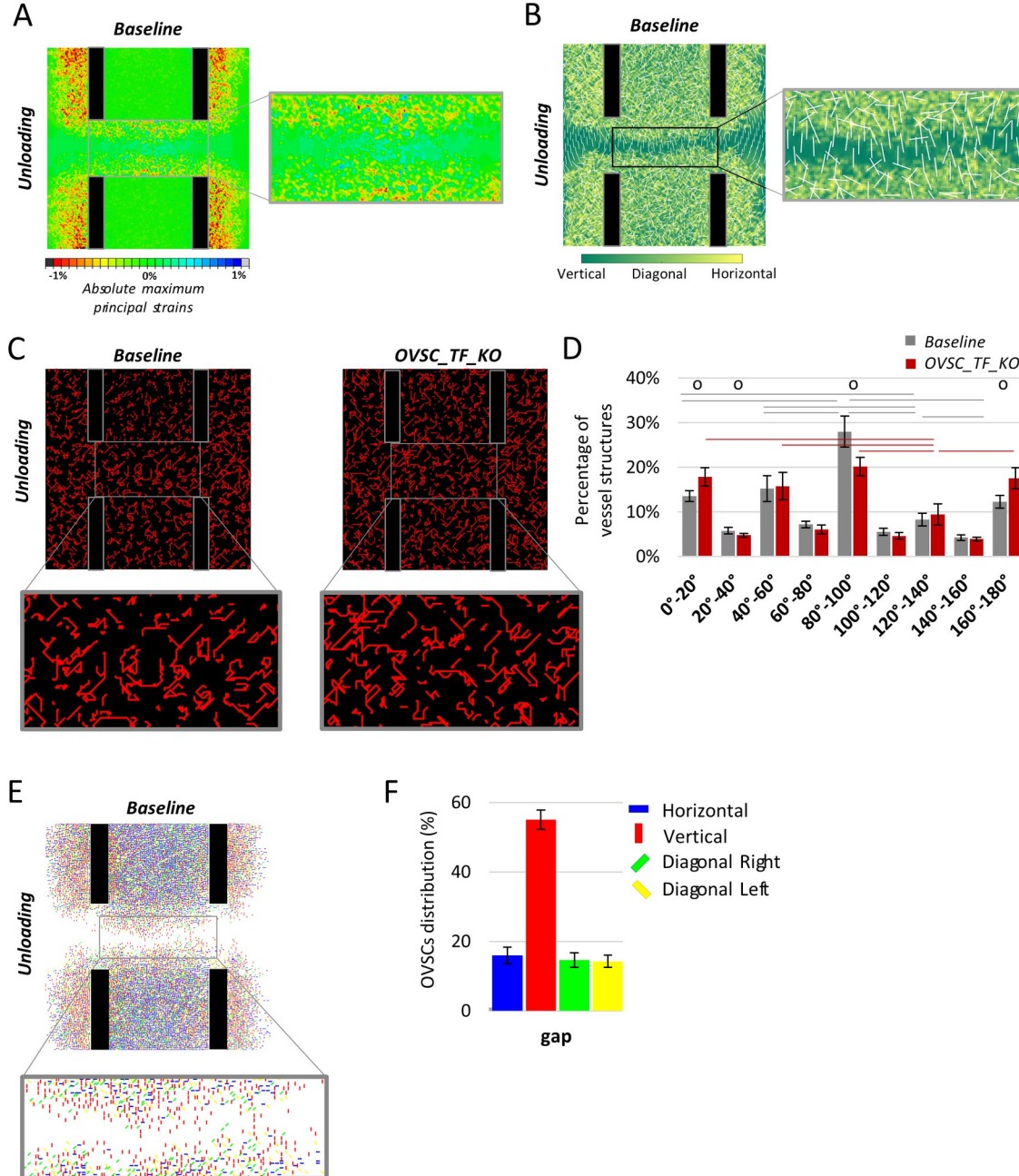

**Fig 8. Predicted vessel and cell organization after unloading the cortices with the baseline model vs. upon stromal cell traction forces inhibition.** (A) Distribution of the absolute maximum principal strains for the unloaded baseline model with a zoom into the gap; (B) Directions of the absolute maximum principal strains for the unloaded baseline model with a zoom into the gap; (C) Predicted vessel distribution for the unloaded baseline model (on the left) and for the unloaded OVSC_TF_KO model (on the right) and zoomed vessel organization in ROI 1; (D) vessel orientation analysis for the unloaded baseline and OVSC_TF_KO models within ROI1. Empty circles indicate a significant difference (p<0.05) between the baseline and OVSC_TF_KO model under unloading. Lines indicate a significant difference in the pairwise comparisons between angle bins in the same model (p<0.05); (E) Predicted OVSCs distribution for the unloaded baseline model and zoomed OVSCs organization within the gap; (F) percentage distribution of OVSCs orientations for the unloaded baseline model within the gap. OVSC_TF_KO = inhibition of OVSCs traction forces.

Interestingly, after removing the high external mechanical signal, the resulting vessel network differed depending on the presence or absence of cell traction forces qualitatively (Fig 8C) and quantitatively (Fig 8D). For both models, the angle bins in which a higher percentage of vessel structures was detected (0˚-20˚,40˚-60˚,80˚-100˚,120˚-140˚,160˚-180˚) were compared statistically. In particular, in the unloaded baseline model (Fig 8C left), the percentage of vessel structures within the gap (ROI1) that took a preferential orientation in the bone long axis direction was significantly higher compared to every other bin (Fig 8D). In the unloaded OVSC_TF_KO model, vessels grew straighter (Fig 8C right) and without any preferential alignment except for a significantly lower percentage of vessels detected in the 120˚-140˚ bin (Fig 8D).

Remarkably, after unloading the baseline model, the sprout preferential alignment in the direction of the bone long axis can be linked to OVSCs orientation within the gap (ROI1) (Fig 8E). Quantitatively, over 50% of OVSCs within the gap (ROI 1) were predicted to align in the direction of the bone long axis (Fig 8F).

## Discussion

Angiogenesis plays a key role in bone regeneration, especially during the early stages of healing. Although it is known that ECs and surrounding OVSCs apply traction forces and respond to the local mechanical environment, the mechanical regulation of the angiogenic process during the early stages of bone healing remains poorly understood. In this study, we investigated the influence of both external loads and cell traction forces on the sprouting process, using a combined *in vivo/in silico* approach. We developed a multiscale computer model of the mechano-regulation of OVSCs and ECs organization within the healing region of a mouse femoral osteotomy under a rigid and semirigid fixation. By comparing model predictions to experimental data, we were able to show that the experimentally observed preferential alignment of sprouts in different areas of the healing region could be explained by the direction and magnitude of principle mechanical strains. In addition, ranges of mechanical strain that have been previously shown to inhibit angiogenesis [66] were found within the gap region under semirigid fixation, and could explain the lack of vessels within this region observed experimentally. Moreover, we predicted a key role of tip EC mechano-response for early sprout organization and suggest that the high *in vivo* external loading can overrule the mechanical communication between OVSCs and ECs during sprouting angiogenesis.

All the mechano-biological rules implemented in the model are based on available experimental observations. For ECs, we put together several studies [17,60–66] to establish a relation between the vessel direction of growth and the magnitude of local mechanical strains around the vessel tip. Under rigid fixation, the level of mechanical strains determined within the gap corresponded to levels previously reported to result in the alignment of ECs perpendicular to the strain direction [17,61–64]. Under semirigid fixation, levels of mechanical strain determined within the gap region were of the order of magnitude of those previously reported to inhibit blood vessel formation [66]. For OVSCs, we modelled their migration towards the healing region as mainly driven by the cell's density gradient during the first 4 days (S6 Fig), simulating an initial attraction of the cells by growth factors, and later biased by *durotaxis* [18]. We show that, after 7 days, OVSCs were predicted to invade the endosteal gap and mainly align parallel to the bone long axis in the periosteal region and perpendicular to the bone cortices in the gap region.

Model predictions were compared to *ex vivo* histology data in terms of vessel density and patterning on the 7th day post-osteotomy. Predicted vessel density nicely matched with the one measured *ex vivo* in different regions of interest under rigid fixation, while the model

overestimated the vessel density within the bone marrow under semirigid fixation. Interestingly, the model was capable of reproducing the lack of vascularity observed experimentally within the endosteal gap under semirigid fixation, which suggests that the lack of vessels can be associated with the high mechanical strains locally.

The predicted preferential alignment of vessels within the three regions of interest was mostly in agreement with the *ex vivo* data under both fixators, although the model failed to capture the preferential alignment observed periosteally (ROI 3) under semirigid fixation. In the rigid fixator scenario, vessels were predicted to align in the direction of the bone long axis within the bone marrow and externally to the cortices, in agreement with vessel patterning observed *ex vivo*. In the marrow and periosteum, this direction coincided with the principal strains direction. Under rigid fixation, vessels were predicted to gradually orient perpendicularly to the bone long axis while invading the endosteal gap, as observed experimentally. This can be explained by the high mechanical strains within the endosteal region and the tendency of vessels to align perpendicular to the strain direction for high levels of mechanical strain [17,61–64]. In the semirigid fixator scenario, the model predicted a preferential alignment of vessels periosteally in the direction perpendicular to the bone long axis as a consequence of the high strains predicted in this region, which was not observed experimentally. Other factors such as haptotactic cues (eg. ECM fibres), not included in the model and known to guide and direct vessel growth, might contribute to the vessel alignment observed in the *ex vivo* images. Indeed, the interaction of cells with matrix topology via contact guidance has been already proven to override the cell re-orientation in response to strains [19,76]. Interestingly, we show here that matrix fibres align parallel to the cortices along the bone periosteum, in line with the neo-vessel preferential alignment observed experimentally in the same region under both fixators. The contact guidance provided by this specific matrix organization at the periosteal site might prevent vessels from re-orienting as a response to high strains. By feeding the model with an experimentally measured fibres distribution and by including contact guidance, *in silico* prediction could better capture vessel patterning at the periosteal site under semirigid fixation, while results would not change under rigid fixation.

The *durotaxis*-based rule guiding OVSCs migration and re-orientation resulted in a preferential cell alignment at day 7 parallel to the cortex surfaces. The distribution of OVSCs orientations was compared to ECM collagen fibres organization during the early healing phase, obtained from a similar mouse osteotomy experiment. In fact, several studies have reported that a bi-directional interaction exists between ECM fibres and cellular alignment [38–42]. The agreement between the predicted OVSCs organization and the experimental ECM fibres alignment suggests that *durotaxis* might contribute to early OVSCs self-organization *in vivo*.

After confirming that the mechano-biological rules implemented can explain ECs and OVSCs organization during early bone healing, we exploited the real power of computational models by performing some *in silico* experiments. The goal was to isolate mechanical signals acting at the cell and tissue level to better understand their role in early sprout patterning, which is not feasible experimentally. When the tip ECs mechano-response was inhibited, vessel fragments grew less organized, shorter and characterized by shape defects as compared to the baseline simulation, suggesting a key role of tip EC mechano-response for vessel growth. Currently, no experimental studies investigated the effect of tip EC mechano-response inhibition during bone regeneration. However, Neto et al. (2018) [77] revealed that the endothelial specific knock-out of YAP-TAZ (key molecules of a mechano-signalling pathway) on the mouse retina leads to a disturbed vessel network formation, similar to our findings. From a clinical perspective, delayed bone healing observed in the elderly has been associated with a reduced cell mechano-response [43]. Here, we show that such an alteration in ECs might lead to impaired early angiogenic response and thus contribute to the clinically observed delayed healing.

Interestingly, our results showed that high external loads can overrule the mechanical communication between OVSCs and ECs. Indeed, after preventing OVSCs from applying traction forces, vessel organization was not affected. From a mechanical perspective, this can be explained by the high strain field created by the external loading conditions, simulating physiological activity, as compared to the small deformation induced locally by cell traction forces. This result might appear to be in contrast with the reported role of stromal cells in supporting the formation of the vascular network [14]. However, *in vitro* experiments of OVSCs-ECs co-cultures available in the literature refer to either uniaxially fixated or free-floating setups, with no external loads [15,16]. Besides, previous studies have demonstrated that cyclic external mechanical loading alone enhanced ECs migration, sprout formation and vessel alignment [78,79]. Further *in vitro* experiments, where inhibition of stromal cell's traction forces would be performed on cyclically loaded substrates, should confirm these observations.

In recent years, clinical studies have shown that unloading hinders bone healing [73,74]. Therefore, the purpose of the unloading scenario was not to replicate a clinically relevant case or suggest a better approach to treat fractures, but to isolate the cell-induced mechanical forces from the high external loads to investigate the cell-cell mechanical communication. After the cell-induced forces were isolated, hints of the ECs-OVSCs mechanical interaction emerged and results are discussed in the context of the available *in vitro* literature. OVSCs oriented themselves mainly along the bone long-axis direction within the gap. Accordingly, vessels aligned following the same direction, driven by the local strain induced by OVSCs contractility. An analogous mechanism of stromal cells-driven organization for carcinoma cells was proposed by Gaggioli and colleagues [80] who showed that stromal fibroblasts pave the way to carcinoma cell migration and invasion through force-mediated matrix remodelling. Despite a preferential alignment of vessels was detected, vessels appeared qualitatively and quantitatively less organized than in the externally loaded simulations, owing to the highly dynamic micro-environment perceived by tip ECs, continuously varying depending on OVSCs orientation and position. Under unloading conditions, the absence of OVSCs traction forces impacted sprout patterning and led to straighter vessels without any preferential orientation. Indeed, as a consequence of OVSCs traction force inhibition, the only strains sensed by tip ECs in the developed model are those generated by themselves along the sprout growth direction. These results are in agreement with the lack of aligned vessel structures obtained by Rosenfeld and colleagues [15] after inhibiting the rho-associated protein kinase (ROCK) in uniaxially-fixated fibrin gels. Taken together, these findings under unloading conditions propose that the strains induced by OVSCs traction forces within the ECM act as a regulator of vessel organization in the absence of high external loads, corroborating previous experimental observations [15,80].

The computer model presented here includes several limitations that need to be mentioned. To save computational time, the model geometry was assumed 2D, despite in reality the healing region is 3D. Nevertheless, the predicted average compressive strain was in agreement with the original 3D model [43] in terms of both magnitude and distribution, confirming the validity of the 2D simplification and specifically the assumptions for loading and boundary conditions. Since a negligible tissue formation was experimentally measured during the first week post-fracture [81], displacement boundary conditions to account for the fixators and the external loading were considered acceptable. In order to apply displacement boundary conditions consistent with the original 3D model developed by Borgiani et al. [43], we assigned identical material properties to the tissues as those reported in that study. However, the granulation tissue mechanical properties remain poorly investigated, especially at the onset of bone healing. Once experimental data are available, a more realistic description of its mechanical behaviour

should be included in the model. A novel approach was established to obtain *ex vivo* section-like images from a simplified 2D ABM. This method ensured getting vessel fragments as the output of the model instead of a continuous vascular network, as observed in *ex vivo* 2D sections. We assumed a constant growth rate for vessels based on *in vivo* measurements [49], although it is known that ECs migration and proliferation rates are influenced by the externally applied load [27] and matrix stiffness [82]. Future work should incorporate a mechanics-dependent vessel growth rate in the computer model, based on the available literature. This could help explain the reduced vascularity observed experimentally under semirigid fixation that the model failed to reproduce within the bone marrow (ROI 2). For OVSCs, no out-of-the-plane migration was included in the model but proliferation and apoptosis rates were scaled from the literature [43] by a factor of 2.25 (18/8) to account for the reduced number of lattice points available in 2D (8 possibilities) as compared to 3D (18 possibilities). For OVSCs, model parameters from fibroblast-like cells were used given the higher amount of data present in the literature on their mechano-regulation. Due to the availability of experimental data, the validation of the model was possible only on day 7. However, the *in silico* model presented in this study is dynamic and allows to view the system at multiple time points which are not sampled experimentally (S1 and S2 Movie files).

We speculate that during early bone healing, ECs may integrate mechanical signals acting at different spatial scales (macroscopic and local) and time scales (minutes for cell traction forces [83] and seconds for stride frequency [84]) in their decision-making process for migration. To correctly capture the effect of mechanical loads acting at different frequencies, it will be necessary to include time-dependent and dissipative behaviours (e.g. viscoelasticity) of tissues in the model. Moreover, ECM fibres (e.g. fibronectin) are known to play a role in the mechanical interaction between cells by providing haptotactic cues for their migration and alignment. In turn, ECM fibres organization is affected by cell traction force-mediated matrix remodelling [41]. Future work will expand the model by including ECM fibre deposition and remodelling. Such a refinement of the model might be able to explain the experimentally observed alignment of vessels within the periosteal region under semirigid fixation (along the bone long-axis) since stromal cells, known to contribute to fibre deposition, were predicted to align in a similar way (along the bone surface).

In conclusion, the levels of external mechanical loads in combination with tip EC mechano-response appear to be essential to enable a proper angiogenic process during early bone healing. Using a multi-scale mechano-biological computer model of ECs and OVSCs and their cellular self-organization processes during the early stages of bone healing, our analyses predicted a good match to *ex vivo* samples from bone healing models in mice. For the first time, we could explain why microvascular networks establish specific patterning in bone healing that comes to a fast and uneventful healing (rigid) or delayed healing (semirigid). Further, our data explain the central role of tip ECs mechano-response within this process and the structural organisation that the micro-vascular network experiences *in vivo*. The knowledge gained in this study could advance the development of regeneration strategies based on the mechanical control of angiogenesis. Devices like stabilization systems and scaffolds should be mechanically optimized to modulate ECs mechano-response and thus promote microvascular invasion at the onset of healing. For example, the stiffness of standard external fixation systems could be adjusted throughout the treatment phase so as to induce principal strain levels and orientations within the healing region that are favourable for sprouting. Tuning sprouting through mechanics at the tissue and cell level would enable a more targeted and controlled therapy approach for the regeneration of bone, without the unwanted side effects that can occur with cell- and growth factor- based treatments.

## Supporting information

**S1 Fig. Schematic of the femoral osteotomy stabilized by a rigid and a semirigid external fixator.** The rigid fixator provides higher stability to the healing region thanks to its homogenous thickness. The semirigid fixator presents a narrowing of its thickness in the central part, leading to reduced stiffness and osteotomy stability.
(TIF)

**S2 Fig. 3D geometry of the mouse osteotomy stabilized with a rigid and a semirigid fixation system.** All the measures are reported in mm. The colour bar identifies the tissues, the material of the external fixator (PEEK) and nails (Titanium).
(TIF)

**S3 Fig. *Ex vivo* vs. *in silico* original bar plots for each region of interest (ROI) under rigid and semirigid fixation.** The percentage of vessel structures is displayed for every 20˚-width bin. *Ex vivo* data are reported as mean ± standard deviation. The ROIs are identified in the bottom.
(TIF)

**S4 Fig. *Ex vivo* vs. *in silico* normalized distribution of vessel orientation for each region of interest (ROI) under rigid and semirigid fixation.** Each pair of curves represents the normalized percentage of vessel structures for every 20˚-width angle bin *ex vivo* vs. *in silico*. The ROIs are identified in the bottom.
(TIF)

**S5 Fig.  (A) Predicted OVSCs self-organization at day 7 for the semirigid fixator; (B) Percentage distribution of OVSCs orientation for the semirigid fixator within the gap and along the periosteum.** On the right, the colour code for cell orientation is reported and the regions of interest are identified.
(TIF)

**S6 Fig. Predicted OVSCs self-organization at day 0 (left) and day 4 (right) for the rigid fixator.** OVSCs migration was modelled as mainly driven by the cell's density gradient during the first 4 days, simulating an initial attraction of the cells by growth factors, which led to a migration of cells towards the core of the endosteal gap without any preferential alignment.
(TIF)

**S1 File. Displacement field due to sprout traction.**
(PDF)

**S2 File. Parameter sweep analysis.**
(PDF)

**S1 Movie. Video of vessels invasion over the first week of bone healing under rigid fixation.**
(GIF)

**S2 Movie. Video of vessels invasion over the first week of bone healing under semirigid fixation.**
(GIF)

## Acknowledgments

We thank Ansgar Petersen for providing the image of the alignment of newly formed collagen fibres in the proximity of the cortex during the initial healing phase obtained via second

harmonic signal, two-photon microscopy. We would like to acknowledge the High-Performance Computing Center North (HLRN) for providing the computational resources that enabled us to conduct the simulations presented in this work.

## Author Contributions

**Conceptualization:** Chiara Dazzi, Holger Gerhardt, Petra Knaus, Georg N. Duda, Sara Checa.

**Funding acquisition:** Holger Gerhardt, Petra Knaus, Georg N. Duda, Sara Checa.

**Investigation:** Chiara Dazzi, Julia Mehl, Mounir Benamar.

**Methodology:** Chiara Dazzi.

**Writing – original draft:** Chiara Dazzi, Sara Checa.

**Writing – review & editing:** Chiara Dazzi, Julia Mehl, Mounir Benamar, Holger Gerhardt, Petra Knaus, Georg N. Duda, Sara Checa.

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
