## [Decision Letter · Decision Letter 0]

15 Aug 2023

Dear Dazzi,

Thank you very much for submitting your manuscript "Extrinsic mechanical loading overrules cell-cell mechanical communication in sprouting angiogenesis during early bone regeneration" for consideration at PLOS Computational Biology.

As with all papers reviewed by the journal, your manuscript was reviewed by members of the editorial board and by several independent reviewers. In light of the reviews (below this email), we would like to invite the resubmission of a significantly-revised version that takes into account the reviewers' comments.

We cannot make any decision about publication until we have seen the revised manuscript and your response to the reviewers' comments. Your revised manuscript is also likely to be sent to reviewers for further evaluation.

Sincerely,

Roeland M.H. Merks, Ph.D

Academic Editor

PLOS Computational Biology

Pedro Mendes

Section Editor

PLOS Computational Biology

Reviewer's Responses to Questions

**Comments to the Authors:**

Reviewer #1: Dazzi et al. develop an in silico model to predict mechano-regulation of sprouting angiogenesis in early bone healing. Computational growth predictions were compared to in vivo findings of a mouse osteotomy model, which was either stabilized with a rigid or a semi-rigid fixation system. Computational modelling demonstrates that mechanical strains and strain directions can explain vessel patterning at the onset of bone healing.

Furthermore, with rigid fixation loss of endothelial tip cell mechano-responsiveness resulted an altered vessel organization, which is agreement with experimental findings on angiogenic processes in other vascular beds.

In particular, the authors show that only in the absence of rigid bone fixation, traction forces generated by the surrounding OVSCs might regulate endothelial sprouting behavior.

Although the computational model has several limitations (which are well discussed) the study shows some interesting findings.

Specific points:

• From a biomechanical perspective the manuscript might in some parts confuse the reader. With a focus on endothelial mechano-regulation on a cellular level, intrinsic loads/forces are generated from within the endothelial cell, while extrinsic loads/forces (on the endothelial cell) are exerted by the surrounded cells (here termed OVSCs). Throughout the manuscript, the authors should try to clarify extrinsic loads on a cellular versus tissue level, e.g. external instead of extrinsic.

• Line 178: Linear elastic properties from previous studies have been utilized to feed the computational model. How have those values been experimentally measured? Other studies (e.g. Ghiasi MS, Chen JE, Rodriguez EK, Vaziri A, Nazarian A. Computational modeling of human bone fracture healing affected by different conditions of initial healing stage. BMC Musculoskelet Disord. 2019 Nov 25;20(1):562. doi: 10.1186/s12891-019-2854-z. PMID: 31767007; PMCID: PMC6878676) demonstrate quite variable elastic modulus values for granulation tissue. How would stiffness values for the tissue affect the here developed model?

• The authors describe and discuss that under unloading conditions and in the absence of OVSC traction forces sprout patterning was impacted and led to straighter vessels without any preferential orientation. However, the (patho)biological consequences of this finding are not discussed.

• Further, the authors claim that upon OVSC traction force inhibition, the only strains sensed by tip ECs are those generated by themselves along the sprout growth direction (Line 516/517). What about the passive matrix/tissue stiffness provided by the presence of the OVSCs? The reviewer believes this statement is simplified from a modelling perspective but does not agree with the biological system.

• External fixation of bone fractures is still the most common treatment strategy. Can the authors explain how mechanical tuning beyond standard fixation could look like?

• There is little to no information on how many independent mouse experiments were performed, quantifications of in vivo/ex vivo data are missing.

• Most of the supplementary figures are lacking figure legends. Those should be added.

• Figure 3D: “By Julia Mehl” should be removed.

• Figure 5C: The figure contains previously published data. The Creative Commons public licenses should be provided.

• Line 482: To isolate “various intrinsic and extrinsic mechanical signals” seems exaggerated and not precise.

Reviewer #2: The review is uploaded as an attachment.

Reviewer #3: General Comments

This manuscript is focused on analyzing the effects of external and internal mechanical cues on neo-vessel growth and orientation. For that purpose, a femoral defect was introduced at each mouse's left femur, and after one week, the mice were euthanized, and different sections at the defect region were collected to observe the angiogenesis pattern. In addition, finite element and agent-based models of the femur defect were created to compare the computational analysis with the experimental outcome. Finally, in the computational model, the effects of the internal mechanical cues (sprout tip mechano-response and stromal cell traction force) and external loading were analyzed by inhibiting each in three different settings.

The models appear to be seeded with random numbers. Is there variation in model results with repeated runs? If so, the authors should quantify and report the variance in their predictions. This would also provide them with the opportunity to perform statistical comparisons with their experimental data, which is currently lacking.

Why was only one histological image compared to model predictions? How much variation was there in the appearance of the histological images between animals?

The authors claim to have similar results from in-silico and ex-vivo setups. However, throughout the manuscript, it is evident that the computational analysis could not predict the experimental results based on vessel growth direction and density for the semirigid fixator group. It raises questions regarding the fidelity of the in-silico model. The lack of agreement in specific cases is likely due to the influence of other queues such as cytokine gradients and ECM orientation. This issue needs to be addressed in places where disagreement was reported.

The finite element model did not completely reflect the experimental setup. For example, the authors used a finite element model with a fracture gap (0.5 mm) different from the experimental subject (0.7 mm). Although they mentioned a gap of 0.5 mm and 0.7 mm, the results would have been more acceptable if the FE model were similar.

The authors could explain why the model could not have a defect similar to the experimental one.

Details regarding the in-silico model are missing in the manuscript. This is crucial for reproducing and validating the results. For example, very little information is given regarding the agent-based model and its coupling with the finite element model.

Specific Comments

Introduction

Page 4: "…external load that is actually transferred to the ECM within the healing region depends on the patient's physical activity but also on the gap size and the mechanical…" – This sentence is difficult to read. Consider rewriting with better clarity.

Page 4: "it remains so far unknown how these different and dynamic mechanical signals impact cell self-organization" – In the same paragraph, it is mentioned how and why stromal cells and neo-vessels reorient due to mechanical signals. So, 'unknown' might be a strong word.

Page 4, Line 87: “… loads-induced…” should be …load-induced…”.

Page 5, Lines 110-111: Suggest citing original papers rather than review (37). This appears to be the first simulation study that included realistic microvessel traction forces:

https://pubmed.ncbi.nlm.nih.gov/25429840/

Materials and Methods

Page 6: "…on the left femur stabilized by a rigid or a semirigid external fixator (MouseExFix 100% and 50%, RISystem)." – The Authors should mention the actual material properties and other parameters used to model the fixator. This will help the reader to better understand the impact of using different fixators.

Page 7: "However, collagen fibres have been reported to relate well with cell orientation" – The authors should provide stronger argument supporting their assumption that the orientation of collagen fibers can be used to identify the orientation of the stromal cells, or identify it as a limitation in the Discussion.

Page 8: "The models were meshed through a regular grid of 4 nodes plane strain elements…" – The authors have not mentioned anything about mesh sensitivity study. This study is needed to ensure the outcomes from computational analysis are not dependent on the size and shape of the mesh elements. The authors also need to discuss the relationship between FE mesh resolution and ABM predictions. The FE mesh was used to define the ABM grid.

Page 8: "Displacement boundary conditions (Ux, Uy, Fig. 1) were applied to the cortical bone fragments which were derived from a previously developed and validated 3D finite element model of the same mouse osteotomy subjected to external physiological loading" – Beside mentioning that the boundary conditions were adopted from a previous paper, the authors should include more details in the manuscript for better understanding and flow of reading.

Page 9: Table 1 – The authors have not mentioned which factors were considered while estimating the values of these parameters. Clarifying this will help investigators in future research works and to reproduce the results.

Page 9, Table 1: The authors should justify or discuss the use of a constant growth rate.

Page 9, Table 1: P1 = P2 = 0.3, was this assumption investigated parametrically?

Page 12: "Specifically, every 15 iterations, (to include a persistence time in a plane (34), all the potential positions surrounding…" – The use of parentheses and commas is confusing. The authors should revise this sentence.

Page 14: "To assess this, we performed in silico experiments where tip ECs did not respond to the local mechanical signals, i.e. P3 was set equal to 0" – The authors did not mention the values of P1 and P2 in this scenario – 0.5?

Page 15: "Specifically, the number of vessels with a high T (T>1.2) and low T (T<=1.2) was compared" – The authors did not explain why the threshold value of 1.2 was chosen.

Page 16: "…under semirigid fixation, the model slightly overestimated vessel density…" – The authors should provide an explanation for why the model underestimated the vessel density at one region and overestimated at other region.

Page 17: Figure 3B – The direction of the strain direction is not clear from the image. The authors should include a color-coded reference to identify the direction from the color.

Page 19: "In silico predictions appeared to match with ex vivo observations in terms of sprouts alignment." – Do the predictions actually match? Although the trend of the bar chart is similar for ex vivo and in silico cases (Figure 4B and 4C), the change in orientation of vessels depending on strain is more prominent for in silico case. The vessel orientation is more random for ex vivo case. The random orientation is also evident in supplemental figure S4, where ROI 2 has almost similar distribution for all 20o bins for both rigid and semirigid cases. In addition, the orientation is opposite in ROI 3 for the semirigid fixator, which the authors failed to explain in their manuscript.

Page 22: "…no remarkable preferential orientation was observed when tip EC mechano-response was removed…" – Although the authors claimed that there was no preferred direction, Figure 6C shows that the percentage of vessels oriented in the 80o – 100o direction higher than all other directions. The authors should also explain the change in preferred direction of the vessels from the baseline case (0o -20o and 160o-180o).

Page 24: "Interestingly, after removing the high extrinsic mechanical signal, the resulting vessel network differed depending on the presence or absence of cell traction forces." – Figure 8C does not imply a significant difference between the two cases. In all 20o bins, the percentage of vessels are similar for baseline and OVSC_TF_KO models.

Discussion

Page 26: "…the model slightly overestimated the vessel density within the endosteal gap and periosteally under semirigid fixation" – The authors mention here that the model overpredicted the vessel density in endosteal gap. However, Figure 3G contradicts their statement, where the in-silico vessel density is lower than experimental observation.

Page 26: "The predicted preferential alignment of vessels was in agreement with the ex vivo data within different regions of interest for both fixators" – This statement does not qualify for all the cases described in this manuscript. As mentioned above, the alignment of vessel for semi rigid fixator was not similar for in silico and ex vivo cases.

Page 27, Lines 477-485: Please rewrite these paragraphs so they are not just 1 or 2 sentences.

Page 27: "Other factors such as haptotactic cues (eg. ECM fibres), not included in the model and known to guide vessel growth, might be responsible for the vessel alignment observed in the ex vivo images." – The authors should shed more light on this topic as the finite element model could replicate the vessel alignment for rigid fixator without the 'other' factors mentioned here. The differences between rigid and semirigid fixator cases should be made clear, which are causing these discrepancies.

The finite element model did not completely reflect the experimental setup. For example, the authors used a finite element model with a fracture gap (0.5 mm) different from the experimental subject (0.7 mm). Although they mentioned a gap of 0.5 mm and 0.7 mm, the results would have been more acceptable if the FE model were similar. The authors should explain why the model did not have a defect that represented the experimental geometry.

**Have the authors made all data and (if applicable) computational code underlying the findings in their manuscript fully available?**

Reviewer #1: Yes

Reviewer #2: Yes

Reviewer #3: Yes

PLOS authors have the option to publish the peer review history of their article (what does this mean?). If published, this will include your full peer review and any attached files.

Reviewer #1: No

Reviewer #2: No

Reviewer #3: No
---

## [Decision Letter · Decision Letter 1]

1 Nov 2023

Dear Prof. Dr. Checa,

We are pleased to inform you that your manuscript 'External mechanical loading overrules cell-cell mechanical communication in sprouting angiogenesis during early bone regeneration' has been provisionally accepted for publication in PLOS Computational Biology.

Best regards,

Roeland M.H. Merks, Ph.D

Academic Editor

PLOS Computational Biology

Pedro Mendes

Section Editor

PLOS Computational Biology

Reviewer's Responses to Questions

**Comments to the Authors:**

Reviewer #1: I appreciate that the authors have addressed all comments and recommend the manuscript for publication.

Reviewer #2: The authors have done an excellent job in revising the manuscript which has greatly improved. They have effectively addressed all my concerns.

Reviewer #3: The authors have done a thorough revision of the manuscript and have addressed all of my prior concerns. Thank you for the detailed, easy to follow review response!

**Have the authors made all data and (if applicable) computational code underlying the findings in their manuscript fully available?**

Reviewer #1: Yes

Reviewer #2: Yes

Reviewer #3: Yes

PLOS authors have the option to publish the peer review history of their article (what does this mean?). If published, this will include your full peer review and any attached files.

Reviewer #1: No

Reviewer #2: No

Reviewer #3: No

---

## [Editor Report · Acceptance letter]

8 Nov 2023

PCOMPBIOL-D-23-00666R1 

External mechanical loading overrules cell-cell mechanical communication in sprouting angiogenesis during early bone regeneration

Dear Dr Checa,

I am pleased to inform you that your manuscript has been formally accepted for publication in PLOS Computational Biology. Your manuscript is now with our production department and you will be notified of the publication date in due course.

With kind regards,

Judit Kozma
